# Polypathologies and Animal Models of Traumatic Brain Injury

**DOI:** 10.3390/brainsci13121709

**Published:** 2023-12-12

**Authors:** Erin Freeman-Jones, William H. Miller, Lorraine M. Work, Josie L. Fullerton

**Affiliations:** 1School of Medicine, Dentistry and Nursing, University of Glasgow, Glasgow G12 8QQ, UK; erinfreemanjones@outlook.com (E.F.-J.); william.h.miller@glasgow.ac.uk (W.H.M.); 2School of Cardiovascular & Metabolic Health, University of Glasgow, Glasgow G12 8TA, UK; josie.fullerton@glasgow.ac.uk

**Keywords:** traumatic brain injury, polypathology, animal models

## Abstract

Traumatic brain injury (TBI) is an important health issue for the worldwide population, as it causes long-term pathological consequences for a diverse group of individuals. We are yet to fully elucidate the significance of TBI polypathologies, such as neuroinflammation and tau hyperphosphorylation, and their contribution to the development of chronic traumatic encephalopathy (CTE) and other neurological conditions. To advance our understanding of TBI, it is necessary to replicate TBI in preclinical models. Commonly used animal models include the weight drop model; these methods model human TBI in various ways and in different animal species. However, animal models have not demonstrated their clinical utility for identifying therapeutic interventions. Many interventions that were successful in improving outcomes for animal models did not translate into clinical benefit for patients. It is important to review current animal models and discuss their strengths and limitations within a TBI context. Modelling human TBI in animals encounters numerous challenges, yet despite these barriers, the TBI research community is working to overcome these difficulties. Developments include advances in biomarkers, standardising, and refining existing models. This progress will improve our ability to model TBI in animals and, therefore, enhance our understanding of TBI and, potentially, how to treat it.

## 1. Introduction

Traumatic brain injury (TBI) is defined as an alteration in brain function, or other evidence of brain pathology, caused by an external force [1]. Awareness surrounding TBI has accelerated, with increasing recognition as a public health challenge that warrants our attention [1]. TBI affects blast-exposed military veterans, professional contact sport athletes, and survivors of domestic violence and road traffic accidents (RTAs) [1,2,3]. Our understanding of the incidence in Scotland estimates a figure of around 445 per 100,000 for men and 195 per 100,000 for women, which is higher than the overall estimate for Europe [4]. Estimating the global burden of TBI requires data from multiple sources and should be interpreted with caution as individuals experiencing TBI in the context of domestic violence or sporting injuries are less likely to seek care for their injury [4].

TBI demonstrates a bimodal incidence pattern with peaks in adolescents aged 15–25 years and older adults aged ≥ 65 years [5]. However, young children aged 0–4 years are at greater risk of mortality from TBI than older children aged 5–14 years [6]. They are also at risk of abusive head trauma (AHT) which is a leading cause of death in this population and is notoriously misdiagnosed by clinicians [7]. In Scotland, the hospital admission rate in children is decreasing, attributed to public health injury prevention methods [4]. However, the incidence of TBI in the elderly is increasing and these individuals are at greater risk for complicated TBI and extended hospitalisation [5]. Our animal models of TBI should highlight age-related differences where possible to allow for more accurate characterisation of the pathology.

TBI exists on a spectrum of severity, with concussion or mild TBI (mTBI) being the most common [8]. Mild TBI is seen as misleading terminology as repeatedly experiencing mTBI can lead to chronic traumatic encephalopathy (CTE) [3]. Whilst CTE is an emerging diagnosis [9], it is recognised as a neurodegenerative disorder associated with repeated mTBI, commonly occurring in contact sports [10]. The terminology has developed over the past century, from Martland’s observations of ‘punch drunk syndrome’ in 1928, ‘dementia pugilistica’ coined in 1937 by Millspaugh, to Critchley’s CTE in 1949 [10]. Regardless of a diagnosis of CTE, TBI is strongly related to neurodegenerative disorders, such as Alzheimer’s disease (AD) [11,12]. Work towards understanding this link is growing and will advance as we deepen our knowledge of TBI itself [12].

Many diseases are complex, but TBI presents an exceptional challenge as the diversity of causes, people affected, and pathology produced make it difficult to model and subsequently treat. Our inability to provide clinical support to this extensive population has frustrated clinicians and scientists alike, as so many promising preclinical successes have failed in the clinical setting [1,13,14]. Neuroprotective treatments, like progesterone and erythropoietin, consistently show promise in animal studies but fail to improve outcomes in clinical trials [13,15]. Our current treatment approach to TBI reflects a logical neuroprotective approach to reduce secondary injury on the brain, through medical and surgical intervention [15]. Nevertheless, researchers continue to explore therapeutic strategies like neurorestoration, with mesenchymal stem cells as an example [15]. Promisingly, data from recent clinical trials suggest that many individuals with moderate or severe TBI regain function over time [16]. These findings can be utilised in a ‘reverse translation’ approach to refine our screening of potential prognostic or therapeutic biomarkers [17].

However, poor bench-to-bedside translation is not specific to TBI itself and we see that other domains of research, like cardiovascular studies, encounter similar challenges [18]. Our efforts to overcome these barriers can be guided by the stroke research community which updated the STAIR guidelines in 2009 to improve consistency in research methods and reporting [19]. Preclinical stroke research has emerged as a leader in the implementation of crucial study design elements such as randomisation and reporting functional outcomes [18,20]. Heterogeneity between preclinical trials remains an issue but the distinct improvement seen in this field indicates a tangible path for TBI to follow.

This review aims to consider our current understanding of the polypathologies of TBI and review the animal models available to illustrate TBI further, whilst acknowledging the various challenges of modelling TBI. Finally, we will explore the discourse around the recent advancements made and whether this has translated into an improved ability to model human TBI in animals.

## 2. Polypathology of TBI

It is necessary to consider TBI as a ‘polypathology’ as we gain an appreciation of how different disease processes form a conglomerate of neuropathology, with a devastating burden on the individual and society [21]. Whilst mTBI and severe TBI (sTBI) are differentiated using the Glasgow Coma Scale (GCS) [22], there is a growing appreciation of the overlap of the neuropathological consequences of repetitive mTBI and a single moderate or severe TBI [21]. A 2019 review of the current pathology observed in mTBI and sTBI suggested the concept of a ‘TBI dose effect’, as multiple mTBIs or a single sTBI could similarly contribute to a threshold that triggers a neurodegenerative response [23]. Our understanding of the neurodegeneration that follows TBI has expanded rapidly in the past few decades. We have identified many mechanisms following the primary mechanical injury and how they interact, as seen in Figure 1.

### 2.1. Diffuse Axonal Injury

One of the most important mechanisms to consider is diffuse axonal injury (DAI). This is a feature of all TBIs, including mTBI, and could represent one of the core substrates that is associated with the development of many secondary injuries [24]. We can identify DAI from histopathology as axonal bulbs, and swellings along axons at the site of injury, which imply failure or disruption of axonal transport [24]. This was evident in autopsy studies of individuals with a moderate–severe TBI, using age-matched controls, demonstrating that this occurs chronically post-TBI, alongside tissue loss in susceptible areas such as the corpus callosum [25]. The evidence for axonal injury developing into chronic axonal neurodegeneration is consistently observed in human sTBI studies and has been observed in gyrencephalic animal models [23]. Whilst axonal injury in mTBI requires further characterisation, mouse models have shown similar axonal pathology after being exposed to repetitive mTBI [8].

### 2.2. Tau, Amyloid, and TAR DNA-Binding Protein-43

Tau is a protein that provides cytoskeletal support for axonal transport and can be phosphorylated physiologically. Tau becomes pathologically hyperphosphorylated in the setting of disrupted axonal transport [26], eventually aggregating in neurofibrillary tangles seen in CTE and AD [27]. Transgenic mice subjected to TBI demonstrated intensive tau pathology compared to sham mice, which spread to other areas with the greatest neural connectivity [27]. This evidence contributed significantly to the emerging field of TBI research, considering tau to spread in a prion-like manner as a mechanistic explanation for the subsequent neurodegeneration observed diffusely across the brain [28]. Similarly, another protein that accumulates pathologically following axonal disruption is amyloid precursor protein (APP) [11], which is commonly used as an indicator for DAI [3]. We see this translated into the accumulation of amyloid beta (Aβ) plaques, hours after TBI, although the mechanism is yet to be fully elucidated [11]. Amyloid beta plaques are seen as less of a hallmark feature of TBI polypathology, as they are observed less consistently in autopsy studies; 30% of single TBI patients who died in the acute phase post-injury demonstrate Aβ accumulation [12]. TAR DNA-binding protein-43 (TDP-43) is a protein involved in RNA processing, implicated in the pathology of many neurodegenerative diseases, such as frontotemporal dementia (FTD) [29]. TDP-43 immunoreactive neuronal cytoplasmic inclusions are mentioned in the 2016 consensus on diagnosing CTE as a ‘frequently seen’ pathology, alongside Aβ plaques, however, not as part of the diagnostic criteria [9]. This suggests that there is further work to understand the role of TDP-43 and Aβ plaques as neuropathological consequences of TBI.

### 2.3. Inflammatory Response to TBI

TBI results in a complex neuroinflammatory response, observed up to 18 years after injury [25]. A primary mechanical injury damages the blood–brain barrier (BBB), cell membranes, and vasculature, as shown in Figure 1. This leads to secondary effects like glutamate and calcium release, mitochondrial dysfunction, and cell death pathway activation, as previously reviewed [30]. The cascade continues with cytokines and chemokines, activation of microglia and astrocytes, and recruitment of peripheral immune cells such as T-cells [31]. This response aids repair and clearance mechanisms, however, it can become harmful and contribute to the subsequent neurodegeneration observed. This is exemplified by the dual role microglia play in healing and repairing, as well as propagating secondary injury and neuronal damage, post-TBI [32].

Autopsy studies have shown widespread diffuse BBB disruption after a moderate to severe TBI, compared to age-matched controls [33]. This was observed in the acute phase, but also following long-term survival from TBI, suggesting that it is a chronic pathological process. This is mirrored in further analysis, linking the absence of the BBB in maintaining homeostasis within the brain to the dysregulation of tau phosphorylation seen [34]. Interestingly, the pattern of BBB dysfunction was shown to be preferential towards the crests of gyri, rather than the depths of sulci [33], which contrasts the tau pathology pattern considered to be pathognomonic of CTE [9,12].

## 3. Current Animal Models of TBI

It is paramount that we attempt to improve the outcomes of those experiencing the polypathologies of TBI. Animal models are essential to understanding the pathological mechanisms behind human TBI. Here we give an overview of the main animal models available, how they induce TBI, and which pathologies can be modelled (Table 1).

### 3.1. Fluid Percussion Injury (FPI) Model

Fluid percussion injury (FPI) is a commonly used and well-characterised model of TBI [35]. Lateral FPI (applied ≥ 3.5 mm lateral to the sagittal suture) has been used more frequently, to study neuronal degeneration and neuroinflammation [30]. The central FPI (midline applied FPI) is used to produce diffuse and concussive injuries [36] and is increasing in use, alongside the interest in sports and blast injuries [37]. A limitation of both FPI models is that the severity of injury produced is inconsistent, and the brainstem involvement corresponds with greater morbidity [36]. Similarly, it has been observed that the FPI and controlled cortical impact (CCI) model dependably produces progressive tissue atrophy in rodents, whereas this finding is not reliably shown in human TBI [14]. However, when used in aged animals, it produces delayed seizures, which mirrors human TBI and posttraumatic epilepsy [5], which is advantageous to the model. FPI models use head constraints and, while this improves reproducibility, it fails to capture the dynamic nature of TBI in humans [38].

### 3.2. Weight Drop Model

Weight drop models peaked during the 1990s [39] and are still commonly used as they are easily operated [36]. Whilst Feeney’s original model involves a craniotomy to produce a mainly focal injury, Shohami’s was designed to replicate human closed head injury (CHI) and Marmarou’s intended to recapitulate vehicle accidents or falls using impact acceleration. Finally, the Maryland model was built on Marmarou’s work to emphasise frontal impact, commonly seen in human TBI in sports or RTA [37]. Previous reviews of this model found significant variation in the weight used and the height of the drop [39] and this model is limited by its association with unintentional skull fractures and rebound injury [36]. Additionally, whilst many preclinical trials focus on reporting neuropathologies [5,39], fewer trials are reporting functional outcomes post-TBI. Only 31% of 335 weight drop publications performed the Neurological Scale Score (NSS), which assesses sensorimotor skills [39]. Our models must remain focused on human-orientated functional outcomes as well as pathology, to be of clinical relevance.

### 3.3. Controlled Cortical Impact (CCI) Model

Controlled cortical impact (CCI) models were developed as part of the early 2000s move away from weight drop towards piston-focused devices [39]. The CCI model is useful for biomechanical studies as it is easy to adjust and measure mechanical parameters, such as velocity, and there is no risk of secondary rebound injury, as seen in gravity-driven devices [36,37]. This method can be used to induce mTBI on an intact skull [36], in the absence of a craniotomy. However, it is important to consider if CCI or FPI models use a craniotomy for their uninjured ‘sham’ mice. Craniotomies have been shown to cause proinflammatory, morphological, and behavioural damage through the surgical procedure alone [40]. It is recommended that these animal models use naïve, anaesthetised animals as a more appropriate control group [41]. Another disadvantage cited of the CCI model is that it does not produce DAI due to the small diameter of the tip delivering the impact and therefore is not a valid model for exploring this significant feature of TBI neuropathology [42]. However, there are models that alter the size of the impactor to improve clinical relevance to DAI [42].

### 3.4. Closed Head Impact Model of Engineered Rotational Acceleration (CHIMERA) Model

In 2014, Namjoshi developed the closed head impact model of engineered rotational acceleration (CHIMERA) to overcome some of the difficulties facing animal models of TBI [38]. Firstly, this model offers precise delivery and analysis of the biomechanical features of the injury, which are critical for enabling comparison, as human TBI is well-characterised for its biomechanical variation [14]. Additionally, this model does not use a craniotomy which increases its strength as an accurate model of human TBI [43] and allows it to be used for multiple behavioural and neuropathological assessments, following unconstrained head impact to a closed skull [38]. However, for ethical reasons and to comply with relevant animal welfare regulations, animal models are commonly anaesthetised or sedated during injury delivery [42]. A systematic review and meta-analysis of trials using anaesthesia in rodents found it to be neuroprotective, with an estimate of improving neurological outcomes by 30–40% [44]. This may influence the findings of neuropathology and behavioural analysis of TBI animal models, making them less translatable to the human clinical setting [43]. This is a key difficulty in replicating the human condition of TBI in animals.

### 3.5. Blast TBI Models

To model blast injuries in animals, various methods have been created and have shown extensive, graded effects of blast injuries on rodents [37]. In 2012, Goldstein showed evidence of CTE-like pathology in mice, 2 weeks after a single controlled blast from a compressed gas-driven shock tube, although this is only using male mice [2]. A challenge of blast models is the frequent involvement of other organ systems, such as the respiratory system, leading to studies using thoracic protective vests to prevent associated morbidity [37,45]. The complexity of current animal blast models poses a barrier to further analysis and comparison to other animal models of TBI [39].

### 3.6. Penetrating TBI Models

Penetrating models of TBI (pTBI) were challenging to develop, due to a high mortality rate associated with the speed of injury delivery [36]. However, using a modified air rifle with a pellet has shown utility in producing histopathological outcomes such as cavitations and gliosis [46]. One of the key advantages of the pTBI is being able to move the injury site, which allows for precise targeting of lesions of the brain [46]. Notably, this model produced a significant increase in reference memory errors during the 7-day testing period using the radial arm maze [46]. However, these findings were produced in a small sample size (*n* = 10 rats) and this model requires further validation. Additionally, this model needs standardisation, as the variation it allows makes the results produced less reproducible [37].

### 3.7. Rotational TBI Models

The majority of animal models used in TBI research are rodents [47] which have lissencephalic brains with relatively minimal white matter. This limits our ability to analyse how their brains respond to injury, as their structural differences mean that some neuropathology, such as white matter degeneration, will not be seen in the model [11]. We know that the large size of the gyrencephalic human brain means that it undergoes greater deformation in response to dynamic or rapid accelerations, compared to smaller, lissencephalic brains responding to similar forces [5,47]. This is significant when we analyse the consequence of TBI, as we aim to mirror the characteristics of the human brain undergoing shear stress as closely as possible.

To account for this, Gennarelli used a rotational acceleration model on gyrencephalic primates, to demonstrate the inertial forces associated with motor vehicle crashes [48]. This model illustrated DAI, prolonged loss of consciousness, and induction of coma in non-human primates [14]. However, the specimens were collected from primates culled at different time points, ranging from 2 h to 8 weeks after injury, which may have had a confounding impact on the results [48]. Moreover, it is not feasible to conduct preclinical studies in large animals on the scale required to investigate TBI, due to financial and ethical implications [14,45]. Therefore, it is imperative to refine rodent models further, as they remain central to preclinical models.

### 3.8. Non-Mammal Models

Alongside the continued use of rodent models, there is a growing appreciation for non-mammal models due to their simplified physiology and cost-efficacy [49].

Drosophila melanogaster flies are used as an animal model that has 70% genetic overlap, a nervous system with glial cells, similar diversity of neurotransmitters, and exhibits most of the behavioural impairment displayed by humans [50]. Using Drosophila flies will augment our ability to explore TBI more efficiently for high-throughput screening of therapeutic compounds, with numerous different models of injury being developed like the omni bead ruptor model for mTBI [49].

Furthermore, zebrafish represent an advantageous model due to their reduced animal husbandry costs and ease of monitoring in batches [51]. A non-surgical closed head injury model in male and female zebrafish produced similar mechanisms of mammalian pathophysiology as well as demonstrating relevant behavioural outcomes to TBI, indicating clear potential for further applications in TBI research [51]. Zebrafish have a shorter lifespan which allows for greater flexibility to use existing weight drop models to investigate the impact of ageing on TBI outcomes [49]. Similarly, Xenopus tadpoles have been shown to demonstrate a wide range of TBI pathologies in response to a focal impact injury model [50]. The variety of models being produced for each non-mammalian species does present a challenge for reducing heterogeneity within the field, as the models have not been characterised as extensively as rodent models of TBI and require further scrutiny as to their utility [49]. Nevertheless, these models will be pivotal to the feasibility of preclinical research using parallel models to improve clinical translation [14].

**Table 1 brainsci-13-01709-t001:** Overview of current animal models of TBI.

Animal Models References	Experimental Procedure	Animals	Technical Features & Variations Used	Pathology	Strengths	Limitations
FPI model [5,14,36,37,42,45]	Fixed animal’s brain is exposed via a craniotomy A cap is attached to the skull and a reservoir of saline water in a cylindrical tube is attached to the cap At the other end of the reservoir there is a transducer measuring pressure changes A pendulum strikes a piston connected to the transducer which conducts a pressure pulse to the dura of the animal This displaces and deforms the brain tissue, with varying severity	Cat Rabbit Swine Rat Dog Mouse	Position of the craniotomy: central-sagittal, lateral-parietal, para-sagittal Can alter height of pendulum to control severity of injury Variations in tube length, material angle	Can cause mild–severe TBI without skull fracture Central: diffuse contusions, haemorrhages, concussion, neuroinflammation, BBB dysfunction Lateral: focal contusions, diffuse subcortical and contralateral injury, haemorrhage	Motor, behavioural, and cognitive deficits seen, lasting for weeks to months EEG abnormalities corresponding to severity of injury Contusions and axonal damage produced in rodents similar to humans Bradycardia, haemorrhage at grey–white matter interface, increased plasma glucose levels, hypertension	High mortality Difficulty calibrating pendulum (improved with addition of microprocessor-controlled pneumatically driven instruments) Requirement of craniotomy Progressive tissue atrophy consistently seen in rodents—unclear if this mirrors human pathophysiology
Weight drop model [14,36,37,45,49,52]	Guided falling of a weight onto the unconstrained skull of an animal: Feeney’s: craniotomy used Marmarou’s: exposed dorsal–ventral skull covered with steel disk resting on foam pad Maryland: impact applied to anterior part of skull Shohami’s: weight applied to one side of unprotected skull resting on hard surface	Rat Mouse Zebrafish	Adjust height of drop Alter the mass, shape, material of weight used With or without craniotomy Change contact surface material or area	Feeney’s: contusion type injury, concussion, traumatic axonal injury, haemorrhage development of a necrotic cavity Shohami’s and Marmarou’s: concussions and traumatic axonal injury primarily, contusions and possible skull fractures Maryland’s: primarily traumatic axonal injury with concussion and haemorrhage	Closely resembles clinical TBI Scalable model as height and mass of weight can be adjusted for severity Simple mechanism and construction Shohami model demonstrates impaired neurological and cognitive outcomes (motor, learning, memory, and anxiety) Use in zebrafish shows genomic changes in CNS injury pathways	Variability seen in injury deliverability Use of metal plate in Marmarou’s/Maryland’s does not reflect human TBI Feeney: craniotomy-associated damage Marmarou’s: higher fatality rate Shohami’s: increased probability of skull fractures
CCI [14,36,37,41,53]	Craniotomy performed on restrained animal skull Use of a pneumatic or electromagnetic impact device to deliver an injury to exposed dura Deformation of underlying cortex	Ferret Mouse Rat Monkey Swine Xenopus	Craniotomy can alter the position and depth. Can alter the speed or angle of the impactor, the diameter of the tip, depth of impact	Cortical tissue loss, subdural haematomas, axonal injury, concussion, BBB dysfunction, increased ICP, haemorrhages if severe Pathology can be focal or diffuse, depending on the severity of the injury delivered	Ability to precisely calibrate injury parameters improves accuracy of injury and therefore reproducibility Reduced risk of rebound injury Motor, emotional, and cognitive deficits seen in walking and memory which correlate with severity and persist for up to 1 year Can be used in small and large animal models	Expensive equipment Craniotomy-associated damage Most CCI models cannot produce DAI Dural laceration as a complication
Penetrating TBI model [14,37,42,46,49]	Animal placed in a frame and head fixed Frontal sinus removed Exposed to different projectiles: missiles, gunshots, sharp objects Creates severe deformational damage through a visible cavity	Cat Dog Monkey Sheep Rat Mouse Zebrafish	Anatomical path, velocity, and angle of projectile Low-velocity pellet model: non-fatal	Model for moderate to severe TBI Creates a large focal cavity in the brain. White and grey matter damage. Brain swelling seizures, neuroinflammation, and BBB dysfunction Extensive intracerebral haemorrhage Low-velocity pellet model produces a cavity, haemorrhage, oedema, gliosis, and white matter degeneration	Cognitive: specifically memory impairment and sensorimotor impairment Neurofunctional deficits correlate with injury severity Produces clinically relevant outcomes like raised ICP Move injury site to target precise lesions	Extensive haemorrhage produced Heat damage from velocity of projectile Less standardised than other models High mortality rate
CHIMERA [14,38,43,54]	Head of animal unconstrained in supine position Pressure-driven piston controlled with a regulator and digital pressure gauge Impact applied to the dorsum of the head, allowing head to flex forward after injury	Rat Mouse Ferret	Can control the parameters of injury including direction, velocity, and impact energy High-speed camera analysis available	Causes axonal injury DAI, neuroinflammation, neurodegeneration, tau, hyperphosphorylation, and white matter inflammation	Non-surgical technique Can be used repeatedly to study long-term effects Semi-automated procedure Variety of dynamic injuries produced Allows for movement of the head after impact Motor, cognitive, and neuropsychiatric outcomes shown with greater consistency than other models	Standardisation of head plates required No large animal model validated for comparison Relatively few publications compared to other models
Primary blast injury [2,14,37,42,45,49]	Animal fixed to metal tube Blast generated through a detonation or release of compressed air	Rat Mouse Pig Drosophila	Head can be restrained or unrestrained Addition of Kevlar vests to protect thorax Amount of explosives or pressure of compressed air used Plastic net to protect from debris or shrapnel	DAI, changes in intracranial pressure, BBB dysfunction, brain oedema, tau hyperphosphorylation, and neuroinflammation	Deficits seen in social recognition, spatial memory, and motor coordination Use of thoracic and abdominal protection minimises mortality Low-level blasts increase ICP and cause cognitive defects Head immobilisation during blast was associated with reduced learning and memory deficits	Model does not accurately recapitulate the dynamic nature of a blast injury Protection from systemic injuries or debris removes the important comorbidities accompanying TBI
Rotational acceleration model [14,36,37,45]	The animal’s head is secured to a device or helmet Induction of a graded rotational acceleration forces	Pig Non-human primate Rabbit	Head restrained or unrestrained Angle of injury, rotation, grading of forces	Primarily DAI Non-human primates: severe TBI, swine: mild to severe TBI with DAI, BBB dysfunction, and damage to the hippocampus	Head rotation is associated with poor functional and histopathological outcomes Highly clinically relevant as a model for falls or collisions	Model is technically sophisticated and expensive Ethical concerns about use of non-human primates

Abbreviations: blood–brain barrier (BBB), controlled cortical impact model (CCI), closed head impact model of engineered rotational acceleration (CHIMERA), diffuse axonal injury (DAI), electroencephalogram (EEG), fluid percussion injury (FPI), intracranial pressure (ICP), penetrating ballistic brain injury (pBBI), traumatic brain injury (TBI).

### 3.9. Outcomes in TBI

One of the key challenges of modelling TBI is our ability to assess the impact of the injury on the animal. We measure the severity of a human TBI traditionally using the Glasgow Coma Scale (GCS) which measures neurological function in three different domains, whereas animal models can be characterised by the severity of the force delivered [55]. Importantly, TBI research increasingly acknowledges the importance of models using clinically relevant functional parameters such as cognitive tests of memory or behavioural measures of anxiety [56]. There are a myriad of tests available, which overlap with stroke research and suffer from the same lack of standardisation in procedure and reporting [20,55].

## 4. Recent In Vivo Advancements

Considerable in vivo advancements have been made in recent years, changing and increasing our ability to model and replicate TBI in animals. This is exemplified by the 2016 Operation Brain Trauma Therapy (OBTT), which advocates for a novel and rigorous approach to a preclinical trial model for TBI, similar to the efforts of the stroke research guidelines in 2009 [19,57]. Implementation of standardisation across multiple models is notable for its efforts to reduce heterogeneity, as well as an appreciation of the necessity of using large gyrencephalic animals and reporting functional outcomes consistently. The inclusion of biomarker assessments allows us to refine the findings of previous studies, showing their potential utility as a prognostic and therapeutic opportunity in human TBI [22].

### 4.1. Biomarkers of TBI

Recent TBI research has highlighted biomarkers as a priority, as they would be notable in their clinical utility in countries with less access to advanced imaging technologies [1,58]. To date, cytoplasmic calcium-binding protein S-100β is the most studied biomarker in the setting of TBI, and there is considerable evidence to support its diagnostic and prognostic capabilities [59]. Additionally, glial fibrillary acidic protein (GFAP) has been studied as a biomarker of injured brain tissue in humans and animals [59]. Recent studies suggest that TSPO may be a reliable prognostic biomarker in human TBI, controlling for other prognostic factors like intracranial pressure [60]. This is mirrored in animal studies suggesting further therapeutic potential through the reduction in apoptosis in male rat models [61].

Increasingly, there is a focus on restoring function in neurodegenerative disease, given the limited time frame for optimal treatment in TBI. Zebrafish demonstrate distinct neuroregenerative capabilities, unlike humans. This represents an exciting opportunity to utilise an animal model’s divergence from human physiology to explore the potential of biomarkers like brain-derived neurotrophic factor (BDNF) to harness restoration following TBI or stroke [62,63]. As we explore the utility of these elements, improvements in administration across the BBB through biomaterials like heparan sulphate may enhance our understanding of the utility of biomarkers and their therapeutic potential [62].

Studies have shown microRNAs (miRNAs), which regulate protein synthesis, could be a useful biomarker in human TBI [43]. These developments may improve our ability to model TBI in animals, as we can validate the expression of certain miRNA changes after a CCI model of injury in mice [64], with overlap between specific miRNAs in humans post-TBI [65]. Studying human and animal model biomarkers in parallel reinforces the importance of viewing preclinical models through the lens of translation into clinical care.

### 4.2. Neuroimaging

Neuroimaging has advanced greatly in recent years and can benefit our ability to model TBI. In 2007, Macdonald showed that diffusion tensor imaging (DTI) could detect DAI in mouse models, unlike conventional magnetic resonance imaging (MRI) [66]. Whilst further research is needed to determine when this imaging should be used in animals and humans for providing prognostic information, it marks a huge step in our ability to assess TBI clinically and preclinically [67]. As we expand our ability to measure human TBI through imaging, this contributes to another new dimension of models—synthetic head models. Researchers have created an idealised gyrencephalic human brain made from polyacrylamide gel and whilst the phantom head has demonstrated early limitations, it has the potential to be refined into a validated model of human TBI; without the use of animals [68].

### 4.3. Genetic Technologies

TBI modelling has been substantially advanced through genetic technologies, allowing us to create transgenic mice, which can highlight certain pathologies [42]. Focusing on tau, genetically engineering mice to express human tau determined how TBI acts as a risk factor for tauopathies such as AD [27]. This enables us to move beyond the constraints of rodent physiology, to recreate TBI pathology with greater precision and accuracy.

Furthermore, studies have investigated how mTBI polypathology is linked to changes in genetic regulation mechanisms in animal models, whilst comparing this to genome-wide association studies (GWASs) of humans [69]. The overlap between genetic changes seen in rodents exposed to TBI and human genes shown to be causally linked to neurodegenerative diseases such as AD suggests potential mechanisms of how TBI creates a predisposition towards these diseases [69].

Using genetic technologies effectively illustrates the complementary roles of large animal and small animal models, as these techniques are much more viable in smaller animals, including non-mammals [70].

Studies have already shown an upregulation of important TBI-mediator genes in Drosophila, but there is potential to build on this with genetic screens for the genes that make them resistant or susceptible to TBI-related pathology, which could allow for therapeutic opportunities [52].

Advancing our understanding of TBI through ‘omics’ such as genomics and transcriptomics is an important consideration but one that remains in its infancy. A recent review notes that one of the limitations is the statistical challenges associated with analysing vast quantities of data from the diverse cohort of TBI-affected individuals, who present with extensive and variable pathology [71]. However, our ability to interpret ‘omics’ data accurately and reliably is improving, and we can continue to use the data to assess the ability of animal models to recapitulate the human response to TBI.

### 4.4. CHIMERA Model

The CHIMERA model is an excellent example of a recent improvement in animal modelling. It primarily produces DAI pathology, which is instrumental to our understanding of human polypathology, and it permits standardisation of the injury parameters, necessary for allowing comparison between laboratories [54]. Whilst the technology itself represents development, preclinical research remains slow to use animals of different genders and ages and this continues to be a barrier to our interpretation of preclinical trial models [54]. However, when we consider the simplicity of the original weight drop model, the CHIMERA animal model is a significant leap forward in our ability to model TBI in animals. This exemplifies the exciting in vivo advancements seen in recent years, as we gain a better appreciation of the complexity of human TBI and the elements we could effectively model in animals [1].

## 5. General Limitations

As we refine rodent models to provide a better, reproducible re-enactment of human TBI, we encounter the same difficulties acknowledged many decades ago. Holburn determined, in 1943, that it is necessary to scale the force of the injury delivery for a smaller brain to fully recapitulate the true effects of human TBI [14]. This involves increasing the size of the inertial forces for a rat brain by 8000%, which is evidently unobtainable. Very few preclinical models discuss how they overcome this distinct barrier to modelling TBI in animals, which calls into question the validity of the models to do so [47].

Further, we understand that TBI is often part of polytrauma, and models should include hypoxia, ischaemia, and potentially substance use, including prescription medications for comorbidities or existing conditions, to provide a more reliable representation of human TBI, as these will exacerbate pathological outcomes [45]. Since this was highlighted in 2005, studies have begun to incorporate these features into CCI models, however, there is still a paucity of studies that utilise this [37]. This demonstrates an area where further advancement is needed.

Furthermore, Bodnar demonstrated that very few studies include aged, female, or young animals [5,39], which suggests that our preclinical models may be of lower utility in evaluating TBI in a large proportion of the population. Evaluation of sex within animal models of TBI presents a largely mixed picture and uncertainty around the specific biological mechanisms underpinning this difference [72]. We can use a lens of ‘mosaicism’ to demonstrate that functional and biological domains are differentially affected in TBI dependent on the sex of the human or animal participant [73]. This furthers the idea of using multiple functional outcomes to evaluate a model, to ensure its relevance to both sexes [73].

Whilst progesterone has not delivered clinical utility in the setting of TBI, its potential neuroprotective qualities were highlighted in trials examining the impact of female sex hormones in TBI outcomes [74]. Our understanding of polypathology and the outcomes associated with TBI will be inherently limited if we exclude female animals from our research.

We know that TBI evolves over years and neuroinflammation is seen to persist for many years [25]. However, a very small proportion of preclinical trials report looking at outcomes beyond 1 month [39], which limits our ability to make valid conclusions about the long-term sequelae of human TBI. Some of these limitations are shared by all models previously discussed and an overview of the barriers to a clinically translatable animal model of human TBI is illustrated below (Figure 2).

## 6. Conclusions

Overall, to evaluate if in vivo advancements have improved our ability to model human TBI in animal models, we must consider the scale of progress that would signify improvement. Given the difficulties facing preclinical trial models, the advancement we have seen is reflective of distinct improvement. Reflection has been a critical element of this: considering the disappointment seen in the progesterone trials [13], the limitations of the animal models [45], and how the TBI research community needs to alter and coordinate its approach to model the condition. It is imperative to highlight the areas where progress has been slower, the minimal use of female rodents in models as an example. As mandated by the National Institute of Health, biological sex must be considered and characterised as part of future TBI study design [75]. Nonetheless, a key theme to consider is how the advancements in different dimensions of TBI research interact with each other. This is emphasised within the previous review of TBI developments, as we observe how the improvements made to biomarkers have a synergistic effect on our ability to use animal models for evaluating TBI [76]. When viewing the evidence base, notable improvements have been made to our ability to model human TBI in animals. Advancement within preclinical stroke research emerged as improvements within clinical trial design, before seeing clinically useful treatments such as nerinetide appear [77]. We can be hopeful that the trajectory of TBI preclinical research will move towards a similar, standardised approach that will ultimately provide therapeutic options to TBI patients.

## Figures and Tables

**Figure 1 brainsci-13-01709-f001:**
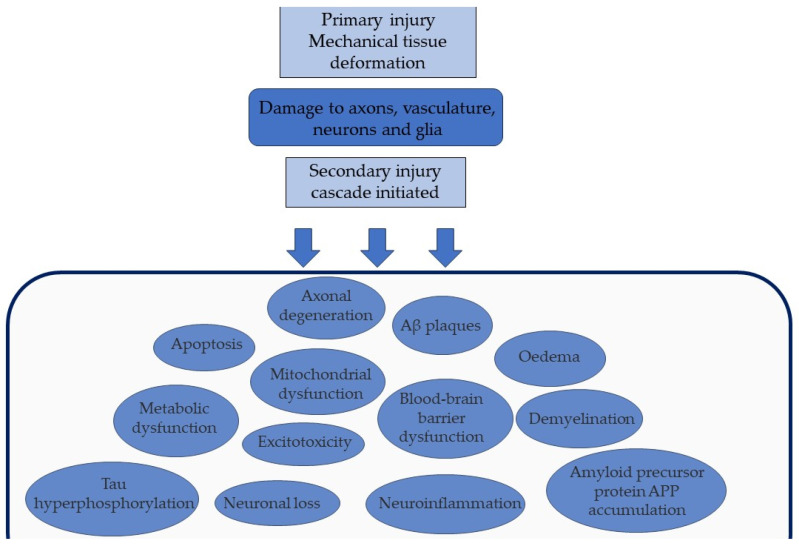
The sequelae of primary and secondary injuries of TBI. The initial primary mechanical injury damages vital structures such as neurons. This expands to secondary injury, a cascade of pathological events including blood–brain barrier dysfunction and excitotoxicity.

**Figure 2 brainsci-13-01709-f002:**
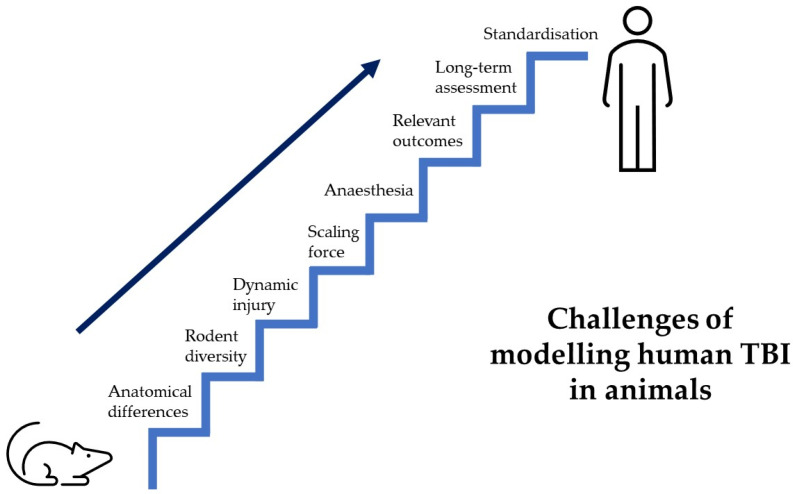
Hurdles of TBI preclinical research. Displays a staircase, representing the many hurdles faced by preclinical trials to create an animal model that recapitulates human TBI.

## Data Availability

Not applicable.

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
