# Peer review of "Polypathologies and Animal Models of Traumatic Brain Injury"

_brainsci, 2023, doi:10.3390/brainsci13121709_

Round 1

Reviewer 1 Report

Comments and Suggestions for Authors

Freeman-Jones et al. reviewed polypathologies and animal models of traumatic brain injury, discussing various animal models and biomarkers for this pathology. The work is interesting, and the manuscript is organized well. I have some minor concerns that need to be addressed before publication.

1.      The introduction should be expanded more, incorporating recent findings on brain injury.

2.      Authors discussed various animal models for this disease, but limited information has been provided. Authors should discuss these modes in detail, incorporating authentication, limitations, and detailed experimental procedures for these models.

3.      Authors should include the translocator protein (TSPO) as a biomarker for brain injury; this protein has been studied extensively as a biomarker for brain injury.

4.      The quality of Figure 2 is not good; the authors should provide a better image for this. 

Comments on the Quality of English Language

Minor editing of English language required

Author Response

REVIEWER #1

We thank the reviewer for their supportive comments around the content and structure of the review. Our efforts to address the minor concerns are detailed here and are highlighted in blue in the resubmitted version of the review.

  1. The introduction should be expanded more, incorporating recent findings on brain injury.

We thank the reviewer for this suggestion and the Introduction now includes regional data, epidemiology, recent findings on those at risk of brain injury, current and emerging treatments for TBI.

  1. Authors discussed various animal models for this disease, but limited information has been provided. Authors should discuss these modes in detail, incorporating authentication, limitations, and detailed experimental procedures for these models.

We have added more detail to the procedures of the animal models in the Table, although for readability we have not been exhaustive about the exact procedures. We have added in several columns to address this important comment meaning that the table now is better placed in landscape format.

  1. Authors should include the translocator protein (TSPO) as a biomarker for brain injury; this protein has been studied extensively as a biomarker for brain injury.

We thank the reviewer for their comment and have added the following wording to the manuscript: “Recent studies suggest that TSPO may be a reliable prognostic biomarker in human TBI controlling for other prognostic factors like intra-cranial pressure [61] This is mirrored in animal studies suggesting further therapeutic potential through reduction in apoptosis in male rat models [62].” and hope that this addresses this point.

  1. The quality of Figure 2 is not good; the authors should provide a better image for this

We recognise that this figure was below publication standard and so we have improved this through changes to the text size, ladder, more generally refined the  figure.

Reviewer 2 Report

Comments and Suggestions for Authors

The authors review the current understanding of polypathologies of TBI and animal models with their strengths and limitations in modeling human TBI. It is comprehensive and good interpreting. There are a few suggestions as follows:

1.     Recent in vivo advancements: According to the recent studies, some researchers also pay attention to the Omics in the field of TBI. The authors could review the Omics contents in this review.

Author Response

REVIEWER #2

We thank the reviewer for their supportive comments around the content and depth of the review. Our efforts to address the minor comment is detailed here and highlighted in blue in the resubmitted version of the review.

  1. Recent in vivo advancements: According to the recent studies, some researchers also pay attention to the Omics in the field of TBI. The authors could review the Omics contents in this review.

We thank the reviewer for this suggestion. Given the time allowed for revision and the length of the manuscript already along with the chosen focus of this review, we have added the following text to the review to point the reader to a recent review which covers Omics in the filed of TBI and trust that this will satisfy the reviewer. Advancing our understanding of TBI through ‘Omics’ such as genomics and transcriptomics, is an important consideration but one that remains in its infancy. A recent review notes that one of the limitations is the statistical challenges associated with analysing vast quantities of data from the diverse cohort of TBI-affected individuals, who present with extensive and variable pathology [72]. However, our ability to interpret ‘Omics’ data accurately and reliably is improving, and we can continue to use the data to assess the ability of animal models to recapitulate the human response to TBI.

Reviewer 3 Report

Comments and Suggestions for Authors

The current review focuses on clinically relevant models against TBI. Importantly, the authors have highlighted various animal models of TBI to improve the prognosis of brain injury. The review is well designed and executed. The studies inclusion is less and must be focused on. Additionally, there are certain aspects which are untouched or needs further modifications. There are few recommendations:

1.The authors can provide recent data on global burden of traumatic brain injury in the introduction section. Is there any regional data available (area-wise)?

2.   The authors didn’t discuss or highlight the age groups of individuals affecting TBI affects individuals and the highly vulnerable ones. Also, determine how age, sex, and genotype affect the course of traumatic brain injury.

3. The authors can briefly include the epidemiology of TBI and current treatments for it.

4. The authors must reframe/work on figures to make them more interesting, reader friendly as these figures seems very basic.

5. Is there any information on the differences between adult and child brain Injuries. Highlight it?

6. There are some similar recent reviews highlighting various animal models of TBI. How is this review different from these? For example:  https://doi.org/10.3390/jcm12123923, https://doi.org/10.1089/neu.2018.6149

7. Clinical aspects and significance are missing. The authors must comment on the importance of clinical trials using TBI biomarkers. Also, the literature review from past and recent studies can be included.

8. It is considered that large animal models provide substantial improvements in clinical translation and better mimic human acute brain injury, there are several drawbacks that impact the ability to use large animal studies. Can we compare the rodent models here? The authors must highlight this aspect.

9. The authors have not gone in depth on range of cognitive tests to evaluate cognitive function applied in experimental models of TBI. They can also discuss the advantages and efficacies of these tests in various animal models.

10. The limitation section can focus and emphasize on models via comparison on existing and need for novel and reliable clinical models.

Author Response

REVIEWER #3

We thank the reviewer for their positive comments around the design and execution of the review article. We detail here changes made to the manuscript to address the recommendations detailed.

  1. The authors can provide recent data on global burden of traumatic brain injury in the introduction section. Is there any regional data available (area-wise)?

We have added text to include regional data for Scotland as per the reviewers recommendation: “Our understanding of the incidence in Scotland estimates a figure of around 445 per 100,000 for men and 195 per 100,000 for women, which is higher than the overall estimate for Europe [4]. Estimating the global burden of TBI requires data from multiple sources and should be interpreted with caution as individuals experiencing TBI in the context of domestic violence or sporting injuries are less likely to seek care for their injury [4].”

  1. The authors didn’t discuss or highlight the age groups of individuals affecting TBI affects individuals and the highly vulnerable ones. Also, determine how age, sex, and genotype affect the course of traumatic brain injury.

We apologise for this oversight on our part relating to these important issues affect TBI. We have amended the manuscript and added further references in support of what we have included. We trust that these changes will be to the satisfaction of the reviewer. They are detailed here and can also be seen in the manuscript: TBI demonstrates a bimodal incidence pattern with peaks in adolescents aged 15-25 years and older adults aged ≥ 65 years [5]. However, young children aged 0-4 years are at greater risk of mortality from TBI than older children aged 5-14 years [6]. They are also at risk of Abusive Head Trauma (AHT) which is a leading cause of death in this population and is notoriously misdiagnosed by clinicians [7]. In Scotland, the hospital admission rate in children is decreasing, attributed to public health injury prevention methods [4]. However, the incidence of TBI in the elderly is increasing and these individuals are at greater risk for complicated TBI and extended hospitalisation [5]. Our animal models of TBI should highlight age-related differences where possible to allow for more accurate characterisation of the pathology.

“Evaluation of sex within animal models of TBI presents a largely mixed picture and uncertainty around the specific biological mechanisms underpinning this difference [73]. We can use a lens of ‘mosaicism’ to demonstrate that functional and biological domains are differentially affected in TBI dependent on the sex of the human or animal participant [74]. This furthers the idea of using multiple functional outcomes to evaluate a model, to ensure its relevance to both sexes [74].”

  1. The authors can briefly include the epidemiology of TBI and current treatments for it.

We thank the reviewer for this suggestion and have incorporated changes around epidemiology as detailed here in points 1 and 2.  For current treatments, we have added the following text to the manuscript: “Neuroprotective treatments, like progesterone and erythropoietin, consistently show promise in animal studies but fail to improve outcomes in clinical trials [13,15]. Our current treatment approach to TBI reflects a logical neuroprotective approach to reduce secondary injury on the brain, through medical and surgical intervention [15]. Never-theless, researchers continue to explore therapeutic strategies like neuro-restoration, with mesenchymal stem cells as an example [15]. Promisingly, data from recent clinical trials suggest that many individuals with moderate or severe TBI regain function over time [16]. These findings can be utilised in a ‘reverse translation’ approach to refine our screening of potential prognostic or therapeutic biomarkers [17].”

  1. The authors must reframe/work on figures to make them more interesting, reader friendly as these figures seems very basic.

We recognise that the figures may not have met the standard needed for publication and so we have improved this through in both figures and also restructured on of these to attempt to make this more reader friendly.

  1. Is there any information on the differences between adult and child brain Injuries. Highlight it?

Please see the changes detailed in relation to point 2 which should also address this recommendation.

  1. There are some similar recent reviews highlighting various animal models of TBI. How is this review different from these?

We thank the reviewer for bringing to our attention these 2 reviews. We might suggest that our review compliments both of those already published – one is focused on biomarkers while the other was published in 2020 and is centred on cognitive tests in 2020. We have referred to and cite each of these reviews in the manuscript to give them due recognition should the reader feel they would like to read more around either of these areas (both of which we cover here).

  1. Clinical aspects and significance are missing. The authors must comment on the importance of clinical trials using TBI biomarkers. Also, the literature review from past and recent studies can be included.

To address this point we have added the following text to the manuscript: “Promisingly, data from recent clinical trials suggest that many individuals with moderate or severe TBI regain function over time [16]. These findings can be utilised in a ‘reverse translation’ approach to refine our screening of potential prognostic or therapeutic biomarkers [17].”

  1. It is considered that large animal models provide substantial improvements in clinical translation and better mimic human acute brain injury, there are several drawbacks that impact the ability to use large animal studies. Can we compare the rodent models here? The authors must highlight this aspect.

We thank the reviewer for this important consideration and we have added the following wording to the manuscript: “Using genetic technologies effectively illustrates the complementary roles of large animal and small animal models, as these techniques are much more viable in smaller animals including non-mammals [71].”

And then also: Moreover, it is not feasible to conduct pre-clinical studies in large animals on the scale required to investigate TBI, due to financial and ethical implications [14,45]. Therefore, it is imperative to refine rodent models further, as they remain central to preclinical models.”

  1. The authors have not gone in depth on range of cognitive tests to evaluate cognitive function applied in experimental models of TBI. They can also discuss the advantages and efficacies of these tests in various animal models.

We thank the reviewer for this recommendation and we have added in a short section to cover outcome measures following the table in the manuscript. Given the depth and breadth of such measures we feel that this is beyond the primary focus of our review and is indeed, covered elsewhere in reviews identified by the reviewer which are now cited in the manuscript. The text added is: One of the key challenges of modelling TBI is our ability to assess the impact of the injury on the animal. We measure the severity of a human TBI traditionally using the Glasgow Coma Scale (GCS) which measures neurological function in 3 different domains, whereas animal models can be characterised by the severity of the force delivered [56]. Importantly, TBI research increasingly acknowledges the importance of models using clinically relevant functional parameters such as cognitive tests of memory or behavioural measures of anxiety [57]. There are a myriad of tests available, which overlap with stroke research, and suffer from the same lack of standardisation in procedure and reporting [20,56].”

  1. The limitation section can focus and emphasize on models via comparison on existing and need for novel and reliable clinical models.

Limitations are described in both the table and in the animal model section. We would refute that instead of novel models being needed that the chimera model and OBTT represent improvements in preclinical models. We believe that a more important consideration is to standardise our existing models and utilise them in combination to achieve better results. Hence, we have not made this changes to address this recommendation.

Reviewer 4 Report

Comments and Suggestions for Authors

The manuscript presented from Erin Freeman-Jones et al., entitled "Polypathologies and animal models of traumatic brain injury" is interesting and original. However there are some critical point to extend:

1) Introduction:

Concerning the different animal models using for TBI, the authors should improve the text including also other vertebrate (fish, lizard, xenopus) and not only mammals, because the aim of the review is to present an extended number of animal models for TBI, in alternative the authors should change title, abstract and text.

2) Penetrating TBI

Indeed for penetrating TBI, zebrafish and other non-mammalian animal models have been used for experiments. I suggest the authors to improve also the table1. 

3) BIOMARKER FOR TBI:

Similar for the paragraph of biomarker, they should mention the role of neurotrophic factors. They should also add new reference.

Suggested reference to add:

- Cacialli P. Neurotrophin time point intervention after traumatic brain injury: from zebrafish to human. IJMS 2021

Zulazmi NA, et al. The utilization of small non-mammals in traumatic brain injury research: A systematic review. CNS Neurosci Ther. 2021

Author Response

REVIEWER #4

We thank the reviewer for recognising the originality and interest of our submitted review. We detail here changes made to the manuscript to address the points detailed.

1) Introduction: Concerning the different animal models using for TBI, the authors should improve the text including also other vertebrate (fish, lizard, xenopus) and not only mammals, because the aim of the review is to present an extended number of animal models for TBI, in alternative the authors should change title, abstract and text.

We thank the reviewer for this point and we have added this to the existing table and also added in a “non-mammal models of TBI” section which we hope is to their satisfaction.

2) Penetrating TBI: Indeed for penetrating TBI, zebrafish and other non-mammalian animal models have been used for experiments. I suggest the authors to improve also the table1.

We agree with the reviewer and have added this detail to the table.

3) BIOMARKER FOR TBI: Similar for the paragraph of biomarker, they should mention the role of neurotrophic factors. They should also add new reference. Suggested reference to add:

- Cacialli P. Neurotrophin time point intervention after traumatic brain injury: from zebrafish to human. IJMS 2021

- Zulazmi NA, et al. The utilization of small non-mammals in traumatic brain injury research: A systematic review. CNS Neurosci Ther. 2021

We thank the reviewer for bringing to our attention these interesting papers. Both have been added in biomarkers section and described.

Round 2

Reviewer 3 Report

Comments and Suggestions for Authors

The authors have done a commendable job of improving the quality of the manuscript. All the necessary comments are taken care of, which further enhanced the scientific vigor and reader-friendly information. Overall, the changes are satisfactory.

Reviewer 4 Report

Comments and Suggestions for Authors

The authors improved the manuscript.